# Silicon Combined with Melatonin Reduces Cd Absorption and Translocation in Maize

**DOI:** 10.3390/plants12203537

**Published:** 2023-10-11

**Authors:** Lina Xu, Xing Xue, Yan Yan, Xiaotong Zhao, Lijie Li, Kun Sheng, Zhiyong Zhang

**Affiliations:** 1College of Agriculture, Henan Institute of Science and Technology, Xinxiang 453003, China; xulina_1023@163.com (L.X.); xx15514124285@163.com (X.X.); yanyan_960916@126.com (Y.Y.); z_haoxiaotong@163.com (X.Z.); lilijie9007@163.com (L.L.); 2School of Hydraulic Engineering, Yellow River Conservancy Technical Institute, Kaifeng 475004, China; 2021821128@yrcti.edu.cn

**Keywords:** melatonin, silicon, cadmium stress, maize, contaminated soil

## Abstract

Cadmium (Cd) is one of the most toxic and widely distributed heavy metal pollutants, posing a huge threat to crop production, food security, and human health. Corn is an important food source and feed crop. Corn growth is subject to Cd stress; thus, reducing cadmium stress, absorption, and transportation is of great significance for achieving high yields, a high efficiency, and sustainable and safe corn production. The use of silicon or melatonin alone can reduce cadmium accumulation and toxicity in plants, but it is unclear whether the combination of silicon and melatonin can further reduce the damage caused by cadmium. Therefore, pot experiments were conducted to study the effects of melatonin and silicon on maize growth and cadmium accumulation. The results showed that cadmium stress significantly inhibited the growth of maize, disrupted its physiological processes, and led to cadmium accumulation in plants. Compared to the single treatment of silicon or melatonin, the combined application of melatonin and silicon significantly alleviated the inhibition of the growth of maize seedlings caused by cadmium stress. This was demonstrated by the increased plant heights, stem diameters, and characteristic root parameters and the bioaccumulation in maize seedlings. Under cadmium stress, the combined application of silicon and melatonin increased the plant height and stem diameter by 17.03% and 59.33%, respectively, and increased the total leaf area by 43.98%. The promotion of corn growth is related to the reduced oxidative damage under cadmium stress, manifested in decreases in the malondialdehyde content and relative conductivity and increases in antioxidant enzyme superoxide dismutase and guaiacol peroxidase activities, as well as in soluble protein and chlorophyll contents. In addition, cadmium accumulation in different parts of maize seedlings and the health risk index of cadmium were significantly reduced, reaching 48.44% (leaves), 19.15% (roots), and 20.86% (health risk index), respectively. Therefore, melatonin and silicon have a significant synergistic effect in inhibiting cadmium absorption and reducing the adverse effects of cadmium toxicity.

## 1. Introduction

Cadmium (Cd) is the third most phytotoxic pollutant after mercury and lead [1]. Cadmium has a high fluidity and hydrophilicity and is easily absorbed by plants. High concentrations of Cd can accumulate in various plant organs [2], causing serious plant toxicity. Cadmium affects physiological and biochemical metabolic homeostasis through metabolic processes such as osmotic stress, oxidative stress responses, membrane tissue disorder, and cytotoxicity, thereby affecting crop growth, development, and yield [3]. Due to the possibility of entering the food chain (including edible plant tissue and grains), Cd can threaten human health [4] and cause organ damage and various diseases such as skeletal deformities, kidney damage, metabolic disorders, and cancer [5]. Therefore, efforts to reduce Cd absorption by plants in contaminated soils or Cd transfer to plant tissues are critical for human food security.

Silicon is a beneficial element that is crucial in regulating the overall physiological and metabolic characteristics of plants, and it plays an important role in enhancing plant resistance to abiotic stress [6]. The alleviating effect of silicon on cadmium toxicity can be attributed to various mechanisms. An appropriate amount of silicon can enhance plant photosynthetic capacities, activate antioxidant enzymes, improve plant growth, and increase the resistance to heavy metal stress [7]. Si can promote cadmium chelation in vacuoles and cadmium fixation in soil, including via the compartmentalization of silicon and metal ions, as well as changes in cell wall structure and composition, thereby affecting cadmium transport between plant organs and reducing heavy metal contents [1,8]. In addition, applying silicon-rich substances to soil can improve the soil’s pH and silicon content and reduce the bioavailability and toxicity of cadmium [9]. Silicon application can reshape the transcriptome of rice roots, indicating that silicon may play a positive role in the whole genome expression regulation of plant resistance to heavy metal/substance stress [10]. Therefore, exogenous silicon application can alleviate the damage caused by cadmium to the morphology and physiological and biochemical processes of plants.

Melatonin (MT) is an indole amine substance widely present in animals and plants. MT can promote the growth and development of plant roots, stems, and explants; regulate the photoperiod and seed germination; and delay leaf aging [11,12]. In addition, MT can enhance the resistance of plants to a range of abiotic stresses, such as drought, waterlogging, high and low temperatures, salt, disease, and heavy metal stress [13,14,15,16,17,18]. Research has found that MT is an effective antioxidant that enhances the ability of plants to scavenge reactive oxygen species, and exogenous MT can induce the synthesis of endogenous MT. Research on tomatoes has shown that exogenous MT significantly enhances plant resistance to Cd stress by enhancing Cd retention in cell walls and vacuoles and the plant’s antioxidant capacity and promoting the plant’s chelation biosynthesis. In addition, limiting the transportation of cadmium from the roots to the buds increases the photochemical efficiency of plant photosystem II to protect photosynthetic processes from cadmium damage [19]. Therefore, MT can serve as a signaling molecule to trigger defense systems, such as antioxidant systems, to alleviate oxidative stress.

Currently, China has more than 2.786 × 10^9^ m^2^ of farmland soil with varying degrees of Cd pollution [20]. Maize (*Zea mays* L.) is a major cereal crop grown worldwide. China’s planting area and total output rank second globally, and agricultural output and quality are key factors in ensuring China’s food security. Maize can accumulate high amounts of Cd in its above-ground components [21,22]. High Cd concentrations can damage the chloroplast tissue, change the membrane permeability, generate reactive oxygen species, cause cell death, destroy plant growth, and ultimately affect maize yields and quality [23]. Thus, to alleviate its toxic effects on maize growth and development and product safety, it has become critical to prevent Cd absorption from the soil and its accumulation in plant organs.

Both MT and Si can promote crop growth, development, and resistance, helping to reduce toxicity and the absorption and accumulation of heavy metals such as cadmium. The combined application of the two increases the tolerance to heavy metal toxicity and achieves better results. Bao et al. reported that the combination of Si and MT can reduce As and Cd migration in rice in polluted soil [24]. Li et al. developed porous silicon (pSi) as a carrier to transport melatonin (MT) to pre-germinated rice seeds, which increased the germination rate under nickel stress by 3.72- and 1.45-fold, respectively, and restored it to 94.97% of the control [25]. Research has found that the combination of 2-HMT and silicon treatment significantly increases ethylene levels, activates antioxidant defense systems, and alleviates chromium stress in *Brassica napus* seedlings. Plant growth or morphological parameters not only have a certain correlation with Cd concentration, but also vary in different plant species, ecological types, and cultivated varieties [26]. There are no reports on combined Si and MT application in maize under cadmium stress. The present study investigated the impact of Cd stress, as well as Cd toxicity, on maize growth and development, while also evaluating the potential benefits of Si and MT treatment, both separately and in combination.

## 2. Results

### 2.1. Effect of MT and Its Compound Application with Si on Corn Growth

Cadmium (Cd) stress significantly inhibited corn seedling growth (Figure 1). Compared with CK, the plant height and stem diameter decreased by 13.81% and 34.89%, respectively, under Cd treatment (Figure 2A,B). Under Cd stress conditions, application of MT and Si alone and in combination significantly increased maize plant height; however, only the combined application significantly increased the stem diameter. Under Cd stress, the application of Si alone, MT alone, or both significantly increased maize plant height and stem diameter. The combined treatment lead to the highest increase in both parameters, with plant height and stem diameter increasing the most (17.03% and 59.33%, respectively), followed by MT alone (10.19% and 50.57%), and Si alone (7.58% and 33.35%) (Figure 1A,B). Furthermore, the combined treatment was found to be significantly more effective at promoting growth than that of Si or MT application alone, which effectively mitigated the inhibitory effects of Cd toxicity on corn seedling growth.

Cadmium (Cd) stress significantly reduced the total leaf area of maize seedlings, which was the lowest under Cd stress (by 43.81% compared to the control). MT, Si, and their combined application increased leaf growth under normal treatment by 10.99%, 19.65%, and 43.98%, respectively. Under Cd stress, the three treatments led to reduced losses in leaf area, which was 27.61%, 28.33%, and 62.55%, respectively, higher than the leaf area under Cd treatment. The combined Si and MT treatment exhibited a better promoting effect on plant growth (Figure 2C).

Biomass is an important indicator of crop growth and material accumulation. Under Cd stress conditions, Si alone and Si + MT significantly increased above-ground and root dry weights compared with the control, but the difference after MT treatment was not significant. Under Cd stress, corn growth was significantly inhibited, resulting in decreases of 34.36% and 24.21% in above-ground and underground biomass, respectively (Figure 2D). Compared with Si or MT alone, the combined treatment achieved better growth improvements, and there was no significant difference compared to Cd-free stress.

### 2.2. Root Characteristics of Corn under Different Treatment Conditions

The root system mainly functions to support the aboveground organs and to absorb and store water and mineral nutrients from the soil. Cd stress significantly inhibited the growth and development of the maize seedling root system, particularly the number of lateral roots. Compared to the control, the number of lateral roots, total root length, root surface area, and main root volume decreased by 40.91%, 35.06%, 23.94%, and 7.27%, respectively, under Cd treatment. The average root diameter was not significantly different to that of the control. Under normal treatment and Cd stress, Si and MT treatment alone, as well as their combination, significantly increased the root length and the number of main and lateral roots. The promoting effect on root surface area, average diameter, and volume under Cd-free stress was the greatest under MT and Si combined treatment. Under Cd stress, the regulation effect of the three treatments on the average root diameter was not significant, and the root surface area and volume were also better promoted under combined treatment (Figure 3A–E). In the absence of Cd stress, compared with the control, Si application alone and combined Si and MT significantly increased the root dry weight, while the difference was not significant under MT treatment. The underground biomass decreased by 24.21% under Cd stress. Si and MT were used together for growth improvement, and there was no significant difference compared to Cd-free conditions (Figure 3F).

### 2.3. Effects of MT, Si, and MT + Si on Photosynthetic Physiology, MDA, and Relative Conductivity of Maize Leaves

Cadmium (Cd) stress significantly reduced the chlorophyll relative value or SPAD and net photosynthetic rate of maize seedling leaves by 14.03% and 40.44%, respectively, compared with the control. Si and MT application alone and in combination significantly increased the SPAD value of maize seedling leaves. The same trend was observed under Cd-free and Cd stress conditions. The net photosynthetic rate reached the maximum under combined treatment without Cd stress, which was significantly higher than that under other treatments. The combined application of Si and MT promoted the relative chlorophyll content and net photosynthetic rate of leaves to a greater extent under Cd stress, followed by treatment with Si and MT separately (Figure 4A,B).

MDA and relative conductivity are important indicators for measuring cell plasma membrane permeability, which indirectly reflects the antioxidant capacity of plant tissues and the degree of plant stress. The MDA content and relative conductivity in maize seedling leaves treated with Cd increased significantly, with increases of 17.24% and 16.96%, respectively, compared with the control (Figure 4C,D). Under Cd-free stress, combined treatment enhanced the plant antioxidant capacity and MDA content significantly decreased compared with the control. There was no significant difference between the other treatments and the control, and the relative leaf conductivity after MT alone, Si alone, and combined treatments was significantly lower than that of the control, but the differences between the three treatments were not significant.

### 2.4. Effects of MT, Si, and MT + Si on Antioxidant Enzymes and Soluble Protein Content in Maize Leaves

Cadmium (Cd) stress led to a significant decrease in SOD and POD in maize leaves, which reached 1.70% and 24.27% of the control levels, respectively. In the absence of Cd stress, the POD content after the three treatments was significantly higher than that of the control treatment, but the SOD content was significantly higher than that of the control treatment only after Si + MT treatment. Under Cd stress, SOD and POD contents in leaves showed an upward trend under Si, MT, and Si + MT. The three treatments significantly increased the SOD and POD contents (Figure 5A,B). The catalase (CAT) content was not significantly different among treatments under normal conditions, and MT + Si significantly increased the CAT content under Cd stress, while the single application of MT or Si had no significant effect (Figure 5C).

Soluble proteins are important osmoregulatory nutrients and indicators of screening resistance. An increase in their content protects the life substance and biofilm formation of cells. Compared to normal conditions, the soluble protein content in corn leaves under Cd stress decreased significantly by 10.41%. Under normal and Cd stress conditions, exogenous MT and Si significantly increased the soluble protein content in leaves. Under Cd stress, the leaf soluble protein content was highest under Si + MT, followed by Si and MT alone. Under no stress, there was no significant difference in soluble protein contents among the three treatments, but they were all significantly higher than that in the control (Figure 5D).

### 2.5. Effects of MT and Its Combined Application with Si on Cd Accumulation and the Maize Seedling Health Risk Index

The Cd content in maize leaves and roots was measured (Figure 6A,B). Under Cd stress, Cd was detected in the leaves and roots. Cd was most easily taken up by plant roots, where its content was much higher than that in the leaves. The application of Si and MT alone and in combination significantly reduced the Cd content in leaves and roots, and the effect of the combined application of Si and MT was the most evident. Cd can accumulate in human organs through the food chain, posing a serious threat to human health. Thus, the health risk index was further calculated, and its trend was the same as that of the Cd content in the leaves and roots (Figure 6C). Compared with Cd stress alone, the Cd content in the leaves and roots and health risk index of all organs under MT and Si composite treatment decreased significantly by 48.44% (leaves), 19.15% (roots), and 20.86% (health risk index) (Figure 6A–C).

### 2.6. Effects of MT and Its Combined Application with Si on Maize Plant and Spike Traits

In the pot experiment, we observed the following observations. Under Cd-free treatment conditions, there was no significant difference in panicle traits. Panicle growth was inhibited under Cd treatment. Si fertilizer, MT, and the combination of the two improved panicle development, promoted panicle grain number, and increased the single panicle grain weight (Figure 7).

Under treatments, the total dry matter mass of stems and leaves was increased by the application of Si and MT alone and in combination. It increased by 6.89%, 11.00%, and 16.02%, respectively, in Cd-free conditions and by 14.69%, 10.40%, and 20.78%, respectively, under cadmium stress (Figure 8A,B). There was no significant difference in the single panicle grain weight under Cd-free treatment, and under Cd stress, the dry weight of single ear grains was promoted by the application of Si and MT alone and in combination, increasing by 20.75%, 35.19%, and 38.02%, respectively, where the combined effect was the best.

### 2.7. Effects of MT and Its Combined Application with Si on Cd Accumulation in Different Plant Organs during Maturity

We measured the Cd content of all parts of maize at the ripening stage, and no Cd was detected in the treated cells under Cd-free conditions. The Cd content in different organs of Cd-treated plants decreased in the order of roots > stems > leaves > bracts > ear shafts > grains, which is closely linked to the operation of these organs. The Cd content in all plant organs in the Cd treatment group was the highest of all treatments. The application of Si fertilizer, MT, and a combination of the two significantly reduced the Cd content, with the combined application of the two having the best effect (Figure 9A–D).

## 3. Discussion

Cadmium is a toxic heavy metal that is receiving increasing attention, as it is one of the main metals that cause grain yield reductions. It has a high solubility and is easily absorbed by and accumulated in plants. Cadmium pollution in agricultural land increases the risk of reduced plant growth and poses a potential threat to health in the food chain [27]. Even if the Cd content in the soil is low (0.3–0.8 mg kg^−1^), it can have toxic effects on crops, such as inhibiting normal cell division and leaf photosynthetic capacity, causing membrane lipid peroxidation, and reducing antioxidant enzyme activities, thereby inhibiting plant growth and reducing yield [3,28]. It is well known that cadmium has a high migration rate in crops and is easily absorbed by the root system, where it is then transported to the upper part and finally enters grains. Therefore, improving cadmium resistance in crops and reducing the cadmium content in edible plant parts are of great significance for ensuring food safety.

Heavy metal accumulation in plants depends on their bioavailability in soil and the physiological characteristics of crops [29]. The survival of plants in polluted environments largely depends on their ability to separate and/or detoxify toxic pollutants such as cadmium. Silicon application has been proven to be a potential technology to control cadmium pollution in contaminated soil. Silicon fertilizers can improve soil quality and alleviate stress on both biological and abiotic crops [3]. Melatonin can regulate antioxidant levels and also enhance heavy metal absorption and chelation to improve heavy metal tolerance [30]. The application of nutrients (Si) or signaling molecules (MT) to reduce heavy metal toxicity in crops is considered an environmentally friendly and feasible method [7,31].

### 3.1. Single or Joint Role of MT and Si in Regulating Maize Growth

The plant morphology, root system, and dry weight are important basic indicators of the resistance to biological or abiotic stress during plant growth and development. Cadmium is a non-essential element in plants, and cadmium stress can lead to slow growth, stunted plants, decreased yields, inhibited root development, and even death in severe cases. This study also showed that cadmium exhibits toxicity to maize plants, hindering plant growth and development under cadmium stress (Figure 1). Growth measurements decreased, including plant height, stem diameter, leaf area, and plant dry weight (Figure 2). Cd also inhibits root growth, manifested as decreases in the number of lateral roots, total length, surface area, and main root volume (Figure 3). This may be attributed to cadmium stress altering the ultrastructure of plant cells, affecting antioxidant stress abilities and inhibiting plant growth by absorbing and utilizing mineral nutrients [29,32]. Research suggests that excessive Cd concentrations affect the biomass and reduce nutrient absorption and Ca, Zn, and Fe concentrations in maize, leading to nutrient deficiency and affecting normal plant growth and development [33].

Exogenous MT treatment can enhance plant resistance to cadmium stress, increase plant biomass (stem length, root length, and dry and fresh weight) [34], and promote the biosynthesis of chlorophyll and plant-chelating peptides [35]. This study shows that under cadmium stress, MT increased the root length, root volume, and lateral root quantity (Figure 3A,D,E). It can be observed that MT plays an important role in regulating plant morphogenesis, especially in the roots. Melatonin can improve the root structure of tomatoes under heavy metal stress [36]. Exogenous MT application can increase the elongation of maize’s main roots, the number of lateral roots, the root surface area and volume, and root abundance [37]. MT can also alleviate Cd-inhibited seedling growth by promoting the growth of above-ground seedlings and underground roots and significantly reduce the Cd toxicity in tobacco plants [38]. It can be observed that melatonin treatment can alleviate the heavy metal-induced inhibition of root growth and vitality.

Silicon is a beneficial element for plants and can improve their growth and development under normal and stressful conditions. Silicon accumulation/transportation occurs from the root to the stem, which has been confirmed in several crops such as rice, corn, and barley (Hordeum vulgare) [8]. In this study, silicon application led to improved plant growth, significantly increased the maize plant height and stem diameter, and increased root length and the number of main and lateral roots. In this study, we found that silicon application increased the silicon content in plants such as wheat, rice, and corn. In addition, silicon can increase the absorption of major and trace elements in plants to alleviate cadmium toxicity [6]. At the same time, silicon can passivate heavy metals in soil and reduce the effective heavy metal content, thereby reducing absorption of heavy metals, promoting plant and root growth, increasing root activity, and improving absorption of nutrients and water in roots [6]. Silicon fertilizer application can improve the growth index and dry weight of rice, as well as the plant height, biomass, and grain dry weight of wheat and corn [7,39].

The application of MT, Si, and MT + Si alleviated the damage caused by Cd to varying degrees. The combined application of MT and Si is most effective at reducing the adverse effects of cadmium stress, promoting plant growth (Figure 1 and Figure 2), increasing stem and leaf dry weight, improving grain traits (Figure 7 and Figure 8), and reducing cadmium accumulation in various organs (Figure 9). These effects were superior to the effects of applying Si or MT alone. Si and MT have promoting effects on plant development and growth, and their combined application exhibits a significant synergistic effect. The same experimental results were also observed in rice, and the combination of Si and MT is much more effective at reducing the absorption and transportation of cadmium and arsenic than using silicon alone, especially in highly polluted soil. This combined effect has also been confirmed to alleviate the adverse effect of other heavy metals. pSi MT nanocomposites have a significant improvement effect on the germination performance of nickel-stressed seeds, and pSi can enhance the relieving effect of melatonin on nickel stress [25]. Combined use can result in overlapping effects, manifested as a stronger resistance, improved plant nutrient absorption, and alleviated growth inhibition.

### 3.2. Single or Joint Role of MT and Si in Alleviating Oxidative Damage

Under cadmium stress, we detected decreases in SPAD values and Pn in maize leaves, increases in MDA content and relative conductivity, and significant decreases in SOD and POD activities (Figure 4 and Figure 5). Both Si and MT treatments and their combined application can improve the abilities of photosynthetic pigments to varying degrees and increase SOD, POD, and CAT contents. The effect of composite application is better than that of single treatment (Figure 3A,B).

One of the immediate reactions of plants under abiotic stress (such as the presence of heavy metals) is the production of reactive oxygen species, which cause severe oxidative damage to the cell structure, organelles, and cell function. Removing reactive oxygen species is the most important defense mechanism in plants to cope with stress conditions [40]. To reduce harm, plants have developed a complex antioxidant system that maintains the internal balance through both enzymatic and non-enzymatic antioxidants. MT is a growth regulator, biostimulant, anti-aging agent, and antioxidant that plays an important role in regulating plant growth and stress resistance, significantly improving plant tolerance to cadmium [41]. It can easily enter cells through the membrane, further entering the nucleus and mitochondria to directly bind to free radicals and reactive oxygen species, thereby maintaining the stability of the plasma membrane [42,43]. Research has found that the melatonin application in hydroponic growth media appears to increase the quantum yield of PSII by 34% [44], while exogenous application of 2-OHMT enhances the growth properties of treated plants, including the photosynthetic rate, the intercellular CO2 concentration, stomatal conductance, and the transpiration rate [45]. Exogenous MT treatment can enhance the antioxidant enzyme activity, increase the antioxidant substance content in plants under cadmium stress, and reduce oxidative damage [34]. MT can protect photosynthetic organs from cadmium damage by limiting the transportation of cadmium from roots to the above-ground plant. Spraying MT on tomato leaves can significantly reduce the cadmium toxicity and reduce the cadmium content in leaves by increasing the antioxidant enzyme activity [19].

Silicon application can significantly reduce the MDA content in wheat seedling leaves under stress, effectively reducing the cadmium toxicity in leaves and protecting the leaves’ physiological functions. Si enhances leaf gas exchange characteristics and chlorophyll contents, reduces oxidative damage and MDA contents, regulates antioxidant enzyme activity to improve reactive oxygen species removal, and improves tolerance to Cd [46]. Si can also regulate reactive oxygen species production and reduce electrolytic leakage [7]. Research has found that silicon application upregulates antioxidant enzyme activity, significantly increasing CAT, SOD, and POD activities in polluted soil. Additionally, it may reduce excessive ROS production under cadmium and arsenic stress conditions [24]. In this experiment, the combined application of MT and Si reduced Cd accumulation in maize, alleviated the damage to leaves, downregulated MDA, improved leaf function, and significantly upregulated the activities of SOD, POD, and CAT, thereby reducing oxidative damage. This effect also is present in rice, and it is speculated that the combined application of Si and MT enhances the tolerance of rice to Cd and As. This may be achieved by increasing the antioxidant enzyme activity and reducing MDA contents to restore the ROS balance and regulate gene expression related to cadmium absorption and transport [24].

### 3.3. Single or Joint Role of MT and Si in Mitigating the Cd Health Risk Index

The food chain is the main pathway for cadmium to enter the human body, which has negative or even fatal effects on human health. The root system is the organ that first comes into contact with heavy metals in the soil and is also the main site for detoxification. Plants absorb and accumulate Cd in the soil through their roots. The Cd content in roots is higher than that in leaves, and different exogenous MT concentrations significantly reduce the Cd concentration in leaves, improving plant tolerance to Cd [43]. In this study, corn without cadmium treatment did not contain detectable cadmium ions. After cadmium treatment, a large amount of cadmium ions accumulated in the roots and above-ground parts of maize seedlings. The Cd content in roots was significantly higher than that in the above-ground parts. After MT application, Cd enrichment in maize seedlings decreased, and exogenous Si also exerted the same effect. When both were used simultaneously, the Cd content was the lowest, the health risk index dropped below 1, and the relief effect was better than the application of either alone. After applying the two exogenous substances, the decrease in Cd content in the upper leaves was greater than that in the roots. Combined MT and Si treatment can promote seedling growth, while also reducing cadmium accumulation in the above-ground and root parts, as well as the health risk index (Figure 6).

MT can stimulate root and stem growth, promoting the production of new roots. In cadmium-contaminated soil, roots exhibit degradation of the plasma membrane and some cell walls, destruction of the cytoplasm, and disorder of the root surface. After spraying with MT, a complete cell structure with no disintegration was observed in the roots. MT application can effectively alleviate cadmium stress in tobacco root cells [38]. Cd competes with mineral nutrients in the same transportation system, altering plasma membrane stability and ion balance [47]. Excessive cadmium accumulation in roots can inhibit the absorption and transportation of nutrients and water, affect plant growth and development, and cause toxicity [46]. Research has shown that these plants may competitively absorb silicon and cadmium. Silicon and cadmium ions compete for exchange sites in soil. Therefore, due to the presence of competing substances, the absorption of cadmium from the soil may decrease [3]. In addition, melatonin appears to alter plant systems not only by directly quenching free radicals, but also by chelating harmful metal ions. Silicon can increase nutrient absorption through the deposition of specific cell walls to enhance plant stress resistance and passivate or co-precipitate heavy metals to reduce their migration and transformation in soil, thereby reducing the cadmium content in roots [47]. Applying silicon fertilizer can reduce the health risk index of cadmium.

The Cd content in various organs in plants was measured at maturity, with the highest Cd content found in the roots, followed by stems, leaves, bracts, cobs, and grains. The application of silicon fertilizer alone, MT alone, and their combination all reduced the Cd content to varying degrees and improved the characteristics of corn ears, with the best effect achieved with the combined application. During heavy metal stress, silicon application has been found to limit the migration of metals [28]. This indicates that reducing root absorption and inhibiting the transfer of cadmium from the root system to the above-ground parts in cadmium-stressed soil are effective measures to reduce cadmium in various organs of maize plants. Silicon can transport heavy metals into areas with inactive metabolism and chelate with them to cause co-precipitation. MT can also promote the chelation of cadmium by cell walls or vacuoles, limit its transportation from roots to stems, inhibit the expression of cadmium-related genes, and minimize its absorption and toxicity in roots. Cai et al. found that the addition of silicon during the jointing stage significantly reduces the biological cadmium concentration and transport factors in above-ground tissues [48]. The combined use of SI and MT can reduce cadmium uptake in plants, which may be due to the regulation of Cd transport efficiency from roots to stems and then to grains by related genes in roots and stems, thereby controlling Cd accumulation in grains [24].

## 4. Materials and Methods

### 4.1. Plant Materials

A new corn variety “Yudan 9953”, bred by Professor Chen Yanhui of Henan Agricultural University, was approved by the National Crop Variety Association in 2018.

### 4.2. Test Method

Maize seeds (Yudan 9953) of uniform shape and size were selected for sowing, with three seeds per pot (one seedling reserved after emergence). Pots were filled with 0.5 kg of soil (10 × 14 cm) in an artificial climate chamber with light for 14 h and darkness for 10 h, a light intensity of 400 μmol/m^−2^s^−1^, a humidity of 50%, and a temperature of 32 °C during the day and 26 °C at night. The soil was treated in four ways: it was sprayed with either 100 mL water; 100 mL 1.59 g/L Na_2_SiO_3_·5H_2_O; 100 mL 0.175 g/L of CdCl_2_; 1.59 g/L Na_2_SiO_3_·5H_2_O; or 0.175 g/L CdCl_2_. After treatment, the soil was placed in a pot and allowed to stand for 24 h. After the seedlings developed three true leaves, they were separated into two groups for treatment. The first group was sprayed with water, while the second group was sprayed with MT. The seedlings were sprayed with 2 mL water or a 100 μmol/L MT solution at 8 PM every night for two consecutive days following the four aforementioned soil treatments. Therefore, there were eight treatments in total in this study: (1) Check (CK): soil treated with 100 mL water and leaves treated with 2 mL water; (2) MT: soil treated with 100 mL water and leaves treated with 2 mL 100 μmol/L MT; (3) Si: soil treated with 100 mL 1.59 g/L Na_2_SiO_3_·5H_2_O and leaves treated with 2 mL water; (4) MT + Si: soil treated with 100 mL 1.59 g/L Na_2_SiO_3_·5H_2_O and leaves treated with 2 mL 100 μmol/L MT; (5) Cd: soil treated with 0.175 g/L CdCl_2_ and leaves treated with 2 mL water; (6) MT + Cd: soil treated with 100 mL 0.175 g/L CdCl_2_ and leaves treated with 2 mL 100 μmol/L MT; (7) Si + Cd: soil treated with 100 mL 1.59 g/L Na_2_SiO_3_·5H_2_O and 0.175 g/L CdCl_2_, and leaves treated with 2 mL water; (8) MT + Si + Cd: soil treated with 100 mL 1.59 g/L Na_2_SiO_3_·5H_2_O and 0.175 g/L CdCl_2_ and leaves treated with 2 mL 100 μmol/L MT. Plants were sampled after 18 days of cadmium treatment, and the roots were washed with deionized water. The roots were collected in order to conduct a root morphological analysis. In addition, leaf samples from each treatment were harvested, immediately frozen in liquid nitrogen, and then stored at −80 °C.

### 4.3. Measurement Index

#### 4.3.1. Morphological Characteristics

Six seedlings exhibiting similar growth patterns were selected from different treatments. After wiping away the surface water, the corn was divided into above-ground and root sections and the biomass was weighed. Firstly, the maximum height of the whole plant was manually measured using a ruler and this was repeated for each of the eight plant treatments. Secondly, the length and width of all fully expanded leaves were measured using a ruler, and the leaf area was calculated (single leaf area of fully expanded leaves = leaf length × maximum leaf width × 0.75). Furthermore, the overall root morphology was studied using an Epson Perfection V850 Pro Photo scanner (Epson America lnc., Long Beach, CA, USA). Subsequently, the total root length, the total root surface area, the total root volume, the average root diameter, and other parameters were analyzed using WinRHIZO 2007 (Regent Instruments Inc., Québec, QC, Canada) and the fresh weight was obtained after scanning the root system. The base of each stem was cut at 3 cm, scanned using an Epson scanner (placed on a flat plane), and the average diameter was analyzed using WinRHIZO 2007 software, with six repetitions for each treatment. The number of lateral roots 10 cm from the base of the main root was determined. Finally, a fresh sample was placed inside a 105 °C oven for 30 min, dried at 80 °C to measure the constant weight, and the above-ground and root system dry weights were obtained.

#### 4.3.2. Net Photosynthetic Rate Determination

The net photosynthetic rate (P_n_) was measured using a portable photosynthetic instrument (Li6800; LI-COR Inc., Lincoln, NE, USA) for five full-spread leaves of the selected plant between 9:00 and 11:00 a.m. The chlorophyll content in five full-spread leaves of the selected plant was measured by using a chlorophyll meter (SPAD-502, Minolta, Tokyo, Japan).

#### 4.3.3. Determination of the Leaf Conductivity

Plant cell membranes play an important role in maintaining the cellular microenvironment and normal metabolism. Under stress conditions, the cell membrane is damaged by electrolyte extravasation, which increases the conductivity of the extract. The third leaf was selected for each treatment. Leaves were rinsed with distilled water, dried, and cut into small pieces. Then, a 0.1 g sample was weighed and placed in a glass test tube. A DDSJ-308A conductivity meter was used to measure the relative conductivity at room temperature between 20 and 25 °C. Each sample was measured three times, and the average was recorded as R1. After measuring the relative conductivity at room temperature, the test tubes were placed in a boiling water bath for 30 min. After cooling to room temperature, the conductivity was measured again. Each sample was measured three times, and the average was recorded as R2. Relative conductivity = (R1/R2) × 100%.

#### 4.3.4. Physiological Indicators

On day 18 after cadmium treatment, about 0.2 g of fresh corn leaves were ground and homogenized with 1 mmol EDTA and 1%PVPP of phosphate buffer extract. A microplate reader (Spark Multi-functional microplate reader) was used to determine the leaf physiology. MDA content was tested as described by Xia et al. [49]. Activities of SOD (EC 1.15.1.1) and POD (EC EC1.11.1.7) were determined by the method of Qiu et al. [50]. The content of soluble protein was tested as reported by Bradford [51].

The MDA content in plant leaves was determined via the thiobarbituric acid colorimetric method. The extract was reacted with a hot water bath (60 °C,15 min) of 0.75% TBA solution containing 10% TCA and then centrifuged. Then, the absorbance of the extract was measured at 450 nm, 532 nm, and 600 nm. The MDA content in the sample was (μmol·g^−1^·Fw) = [MDA concentration in the extract (μmol/mL) × total amount of the extract (mL)/sample fresh weight (g)]. The enzyme activity was calculated as U/mg protein based on the measured soluble protein.

Superoxide dismutase (SOD; EC1.15.1.1): Riboflavin is prone to oxidation under light and aerobic conditions to produce superoxide ions, thus reducing NBT and resulting in a blue color with a maximum absorption peak at 560 nm. The SOD enzyme is an oxygen-radical scavenger, and the enzyme activity in plants is indicated by the degree of reaction. The SOD enzyme activity unit (U) was measured via the nitroblue tetrazolium (NBT) photochemical reduction method and represented as the NBT photoreduction inhibition by 50%.

Peroxidase (POD; EC1.11.1.7): The peroxidase extract reacted with guaiacol and hydrogen peroxide to produce a brown solution with a maximum absorption luminosity at 475 nm. The guaiacol colorimetric method was used to measure the POD activity in the plant shoots, with an increase of 0.01 in the absorbance value considered the enzyme activity unit. It was measured rapidly under the catalytic reaction of hydrogen peroxide and once every 15 s for a total of 5 min. The POD enzyme activity (U/mg·FW·min) was calculated based on the reaction curve.

The soluble protein content was determined with the Coomassie brilliant blue G-250 method; 30 μL of extract and 170 μL of Coomassie bright blue G-250 solution were mixed and then left for 5 min, with the measured absorbance value at 595 nm (minus the control) used in the standard curve equation to calculate the protein content in the reaction system.

#### 4.3.5. Cd Content and Health Risk Index

All parts of the plant were ground, crushed, and weighed to obtain the dry weight. Then, 0.05 g of sample was added into a tetrafluoro crucible, 20 mL of nitric acid: perchloric acid (3:1) was added for nitration for 12 h, and the tetrafluoro crucible was placed onto an electric heating plate (at about 170 °C) in a fume hood to remove the acid [52]. The reaction was stopped when the volume of the solution became the size of a soybean, and the solution was subsequently cooled, fixed at a 25 mL volume with 3% nitric acid, shaken well, and stored for further use. Finally, the Cd content was determined using an Optima 2100 DV inductively coupled plasma emission spectrometer from Perkin Elmer (PE).

The health risk index [2] was calculated as follows: health risk index = C (CD) × coefficient (C) × daily food intake (DFI)/average body weight (ABW) × oral reference dose of cadmium (Ordc). Here, C (CD) is the Cd content (mg/kg); C is the correction coefficient, with a value of 0.085; and DFI is 0.4 kg/person/d according to the provisional standard proposed by Food and Agriculture Organization of the United Nations (FAO)/World Health Organization (WHO). The adult ABW was set to 70 kg and Ordc was 0.001 mg/kg/day, according to the United States Environmental Protection Agency (US EPA) (1985).

### 4.4. Data Analysis and Statistics

Excel 2016 was used to process and plot the experimental data, and Statistical Product and Service Solutions (SPSS) (SPSS Inc., Chicago, IL, USA) was used for analysis of variance.

## 5. Conclusions

The results indicate that cadmium stress damages cell membranes, hinders the growth of above-ground and root plant parts, and exposes plants to oxidative stress. The combined application of MT and Si has a synergistic effect, reducing the cadmium content in various parts of maize under cadmium stress. Compared to MT or Si application alone, the combination of the two is more beneficial for maize resistance to Cd stress by reducing Cd absorption and oxidative stress, enhancing antioxidant enzyme activity, and maintaining higher photosynthesis rates and improved growth, resulting in a lower Cd health risk index (Figure 10). Our study investigated the changes in above-ground and root architectures, plant photosynthetic physiological processes, and cadmium uptake regulation in various organs of maize under cadmium stress under the combined application of MT and Si. The experimental results demonstrate that this application is an eco-friendly and effective method for reducing cadmium toxicity in maize. We have limited information on the mechanism through which MT and Si application alleviates cadmium stress. In the future, we will further clarify the processes of Cd chelation, absorption, and transport in terms of related gene regulation.

## Figures and Tables

**Figure 1 plants-12-03537-f001:**
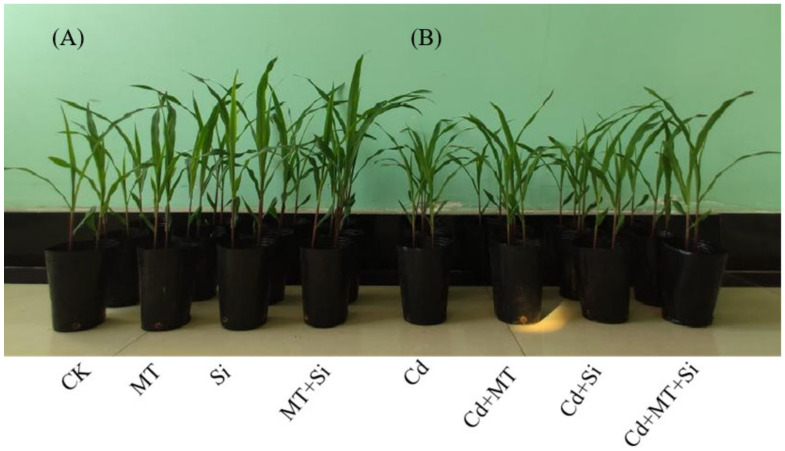
Effects of cadmium (Cd), melatonin (MT), silicon (Si), and MT + Si on maize seedling growth. Check (CK).

**Figure 2 plants-12-03537-f002:**
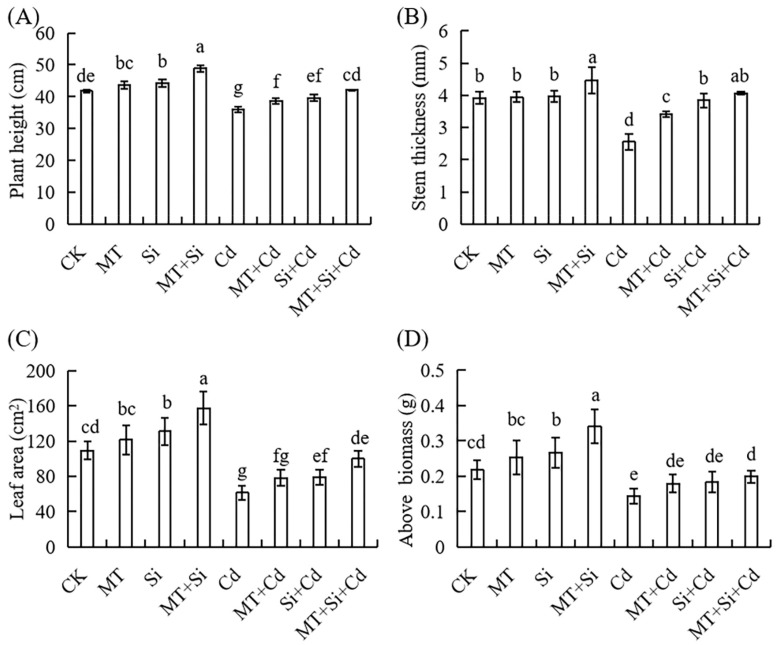
Effects of Cd, MT, Si, and MT + Si on plant height (**A**), stem diameter (**B**), leaf area (**C**), and aboveground dry weight (**D**). Data are shown as means ± standard error (SE) (n = 5). Different letters represent significant differences among treatments (Duncan’s multiple range test, *p* < 0.05).

**Figure 3 plants-12-03537-f003:**
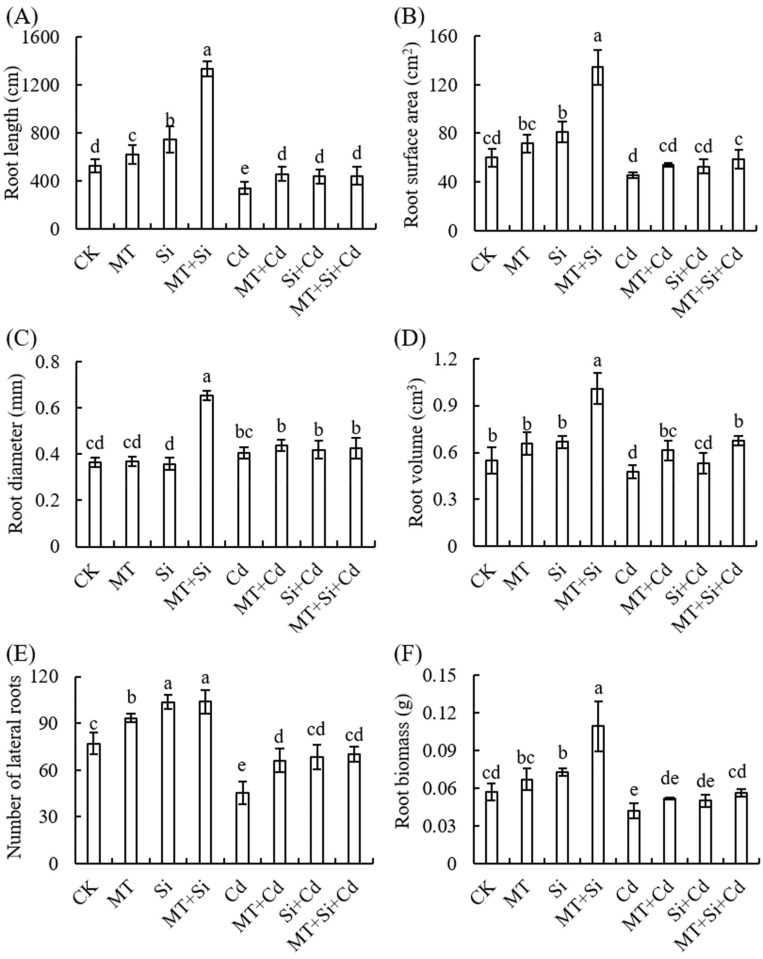
Effect of Cd, MT, Si, and MT + Si combination on root maize characteristics (**A**–**F**). Data are shown as means ± SE (n = 4). Different letters represent significant differences among treatments (Duncan’s multiple range test, *p* < 0.05).

**Figure 4 plants-12-03537-f004:**
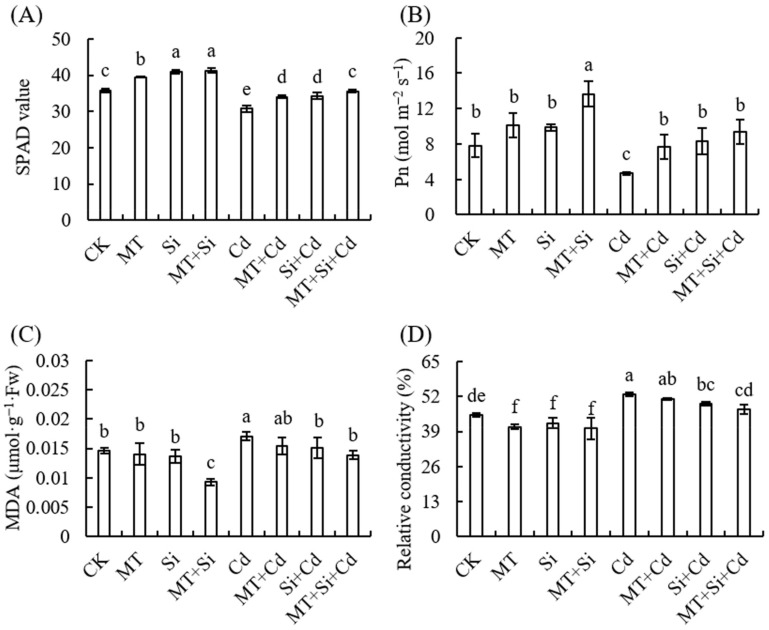
Effects of Cd, MT, Si, and MT + Si combination on chlorophyll content (**A**), net photosynthetic rate (**B**), malondialdehyde (**C**), and relative conductivity of maize leaves (**D**). Data are shown as means ± SE (n = 4). Different letters represent significant differences among treatments (Duncan’s multiple range test, *p* < 0.05).

**Figure 5 plants-12-03537-f005:**
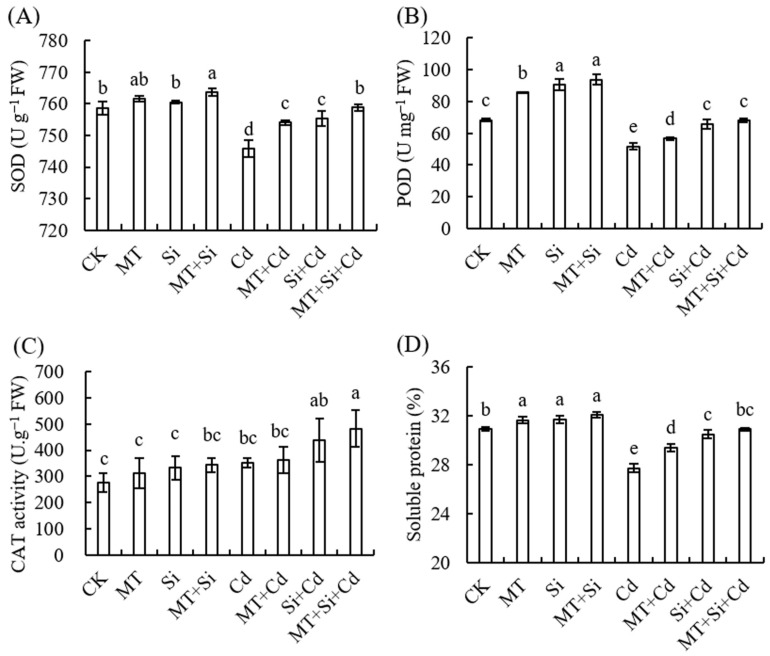
Effect of Cd, MT, Si, and MT + Si on antioxidant enzymes superoxide dismutase (SOD) (**A**), peroxidase (POD) (**B**), catalase (CAT) (**C**), and soluble protein content in maize leaves (**D**). Data are shown as means ± SE (n = 4). Different letters represent significant differences among treatments (Duncan’s multiple range test, *p* < 0.05).

**Figure 6 plants-12-03537-f006:**
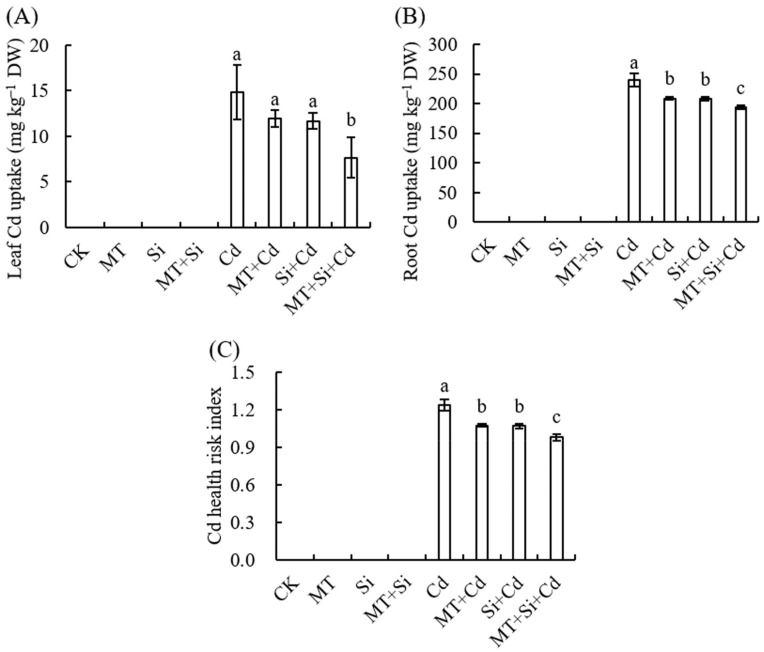
Effects of MT, Si, and MT + Si on Cd content in leaves (**A**), Cd content in roots (**B**), and health risk index (**C**) of maize. Data are shown as means ± SE (n = 4) for dry weight (DW) of leaves (**A**) and roots (**B**). Different letters represent significant differences among treatments (Duncan’s multiple range test, *p* < 0.05).

**Figure 7 plants-12-03537-f007:**
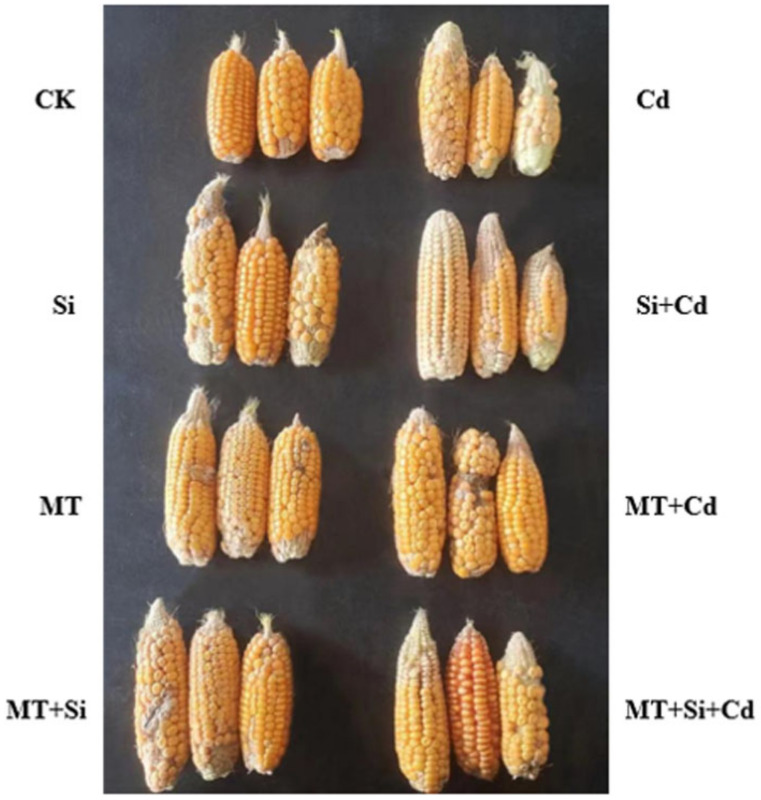
Effects of Cd, MT, Si, and MT + Si on the spike trait.

**Figure 8 plants-12-03537-f008:**
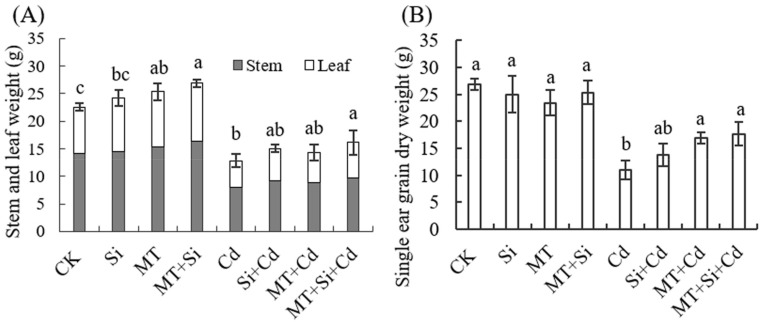
Effects of Cd, MT, Si, and MT + Si on stem and leaf dry weight (**A**) and single ear grain dry weight (**B**). Data are shown as means ± SE (n = 4). Different letters represent significant differences among treatments (Duncan’s multiple range test, *p* < 0.05).

**Figure 9 plants-12-03537-f009:**
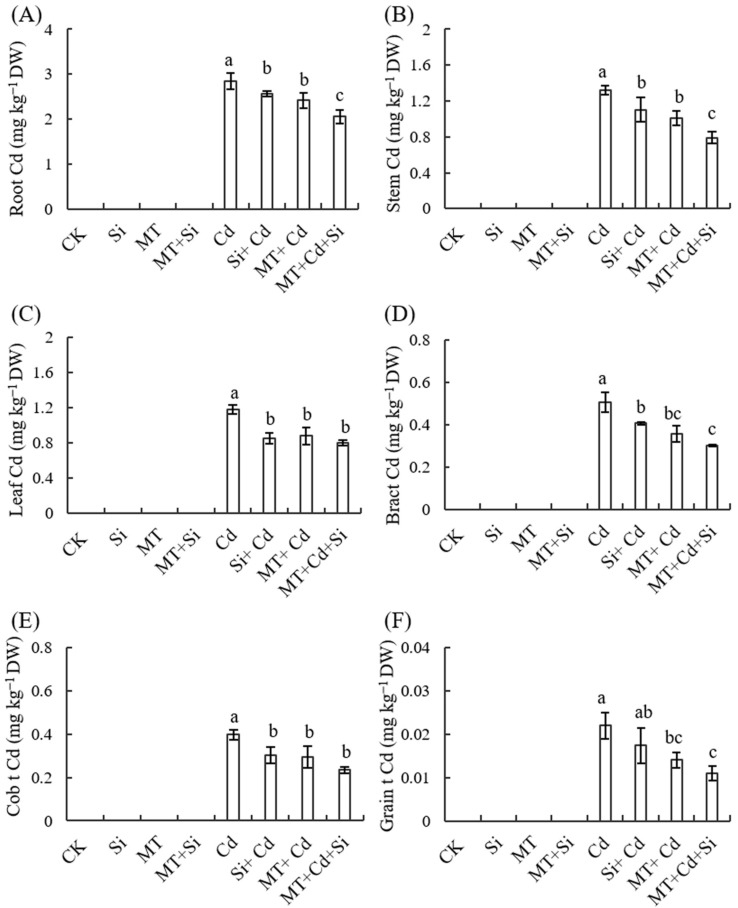
Effects of Cd, MT, Si, and MT + Si on Cd contents in roots (**A**), stems (**B**), leaves (**C**), bracts (**D**), cobs (**E**), and grains (**F**). Data are shown as means ± SE (n = 4). Different letters represent significant differences among treatments (Duncan’s multiple range test, *p* < 0.05).

**Figure 10 plants-12-03537-f010:**
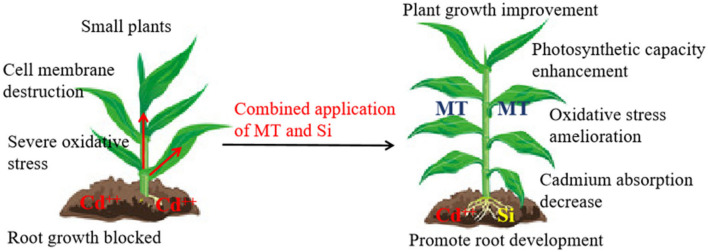
The joint application of MT and Si can alleviate the degree of oxidative stress in maize under Cd stress and promote above-ground and root growth.

## Data Availability

Data will be made available on request.

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
