# Peer review of "Silicon Combined with Melatonin Reduces Cd Absorption and Translocation in Maize"

_plants, 2023, doi:10.3390/plants12203537_

Round 1

Reviewer 1 Report

There are some aspects that have to be improved:

1.      Introduction: The introduction does a good job of explaining the individual roles of Cd, Si, and MT in plant physiology and their importance. However, it could be strengthened by explicitly stating the specific hypothesis or question that this study aims to address.

2.      While the introduction provides a lot of valuable information, it could be made more concise by eliminating redundant phrases or sentences.

3.     The section after Introduction should be Materials and Methods, rather than Results.

4.      Discussion:

-        Please check your format and correct your subsection number in Line 288, 332, 369

-        Line 264-287: could be more concise to summarize the key points briefly before delving into the specific findings.

-        The section mentions mechanisms by which MT and Si alleviate Cd toxicity, such as reducing oxidative stress and Cd accumulation. While these are briefly mentioned, consider providing a more in-depth explanation of these mechanisms and their relevance to the study's findings.

-        Significance for Food Safety: Emphasize the importance of your findings for food safety. Discuss how the reduction in Cd content in edible plant parts can contribute to safer and healthier food production. Make a clear connection between your research and its potential real-world applications.

-        Consider including a brief section on potential future research directions. Are there any unanswered questions or areas for further exploration that your study has revealed? This can help inspire future research in this field.

Minor editing of English language required

Reviewer 2 Report

I have read this article titled (Mitigation of Cadmium Accumulation and Toxicity in Maize (Zea mays L.) by Joint Application of Melatonin and Silicon), its well written bt I have a major suggestions and question's.

1- In line 72-75, I dont agree with that. There are many studies published in the combination of Cd and MT as follows:

https://www.sciencedirect.com/science/article/pii/S0301479721004059

https://www.sciencedirect.com/science/article/pii/S030442382300451X

https://pubs.rsc.org/en/content/articlelanding/2022/en/d2en00187j

https://www.mdpi.com/1420-3049/26/4/862

So, the authors must clear the new results and new goal differ from these studies. because at this case there are any new results or any novelty.

2- The discussion must be updated with published articles last three years 2021 to 2023.

3- In line 275-275. Here the authors mentioned there have been few studies, I am not agree with that also. As I commented before, the authors should be discuss the previous studies results and clear his new results and what are the differences from the published studies. And here authors must cite which few studies they means.

4- In line 474-482. The authors measured the enzymes activity and soluble protein, why they didn't calculated the enzymes activity as U/mg protein.min. I suggest to change the units as  U/mg protein.min.

5- In conclusion line 5222-523, Its not new strategy, re-write the conclusion with the new results differ from the published studies and add the applications and future work in this area.

6- The references must be updated with published articles last three years 2021 to 2023 in discussion, introduction and conclusion.

Reviewer 3 Report

The manuscript entitled “Mitigation of Cadmium Accumulation and Toxicity in Maize (Zea mays L.) by Joint Application of Melatonin and Silicon” showed how melatonin and silicon application  can be beneficial under cadmium stress. Here are some comments on the manuscript.

The manuscript is written carelessly. Abstract doesn’t reflect the results of the present study. In materials and methods section the unit and other should be rewritten according to journal format. Presentation of data in results section is poor. Discussion section is poor, explanation of the parameters changes due to Cd stress and mitigatory effects of melatonin and Si should be discussed properly.

Title: “Mitigation of Cadmium Accumulation and Toxicity in Maize (Zea mays L.) by Joint Application of Melatonin and Silicon”

The cadmium accumulation cannot be mitigated. The accumulation can be reduced. Please revise this.

Abstract:

Line 8: “Zea mays”. Should be italic.

Line 10: “Silicon (Si)” instead of “Silicon”

Line 12: “on growth and Cd accumulation of Cd stressed maize plants” instead of “on maize growth and Cd accumulation”.

Line 13: Please mention the disrupted physiological processes under Cd stress.

Line 14: “improved growth of Cd stressed maize seedling” instead of “alleviated Cd stress-inhibited maize seedling growth”

Line 14: “which is understood by improvements in various seedling characteristics” instead of “evident by im-”

Line 16: “relative conductivity” not similar to Fig. 4D. The term should be similar throughout the whole manuscript.

Line 17-18: “Furthermore, Cd health risk and enrichment” – not understood. Rewrite please.

Keywords: Keywords should be different from the words of the title.

Introduction

Line 24-29: Effect of Cd on plants should be discussed broadly. 

Materials and methods:

1.     No citation or references are mentioned for the methodology of the following parameters:

Net photosynthetic rate (Pn)

Soil Plant Analysis Development (SPAD)

Malondialdehyde (MDA)

Superoxide dismutase (SOD)

Peroxidase (POD)

Soluble protein content

Cd content

2.     Soil Plant Analysis Development (SPAD) reading is not Chlorophyll content. Please revise.

3.     Please include more information for the following parameters:

Net photosynthetic rate (Pn)

Soil Plant Analysis Development (SPAD)

Malondialdehyde (MDA)

Superoxide dismutase (SOD)

Peroxidase (POD)

Soluble protein content

Results:

Is there 8 treatments? So, why those data were presented in two separate graphs.

Pease write all units following SI method and journal format. e.g. mg kg-1 instead of mg/kg. Please fix all others too.

Line 164: Fig 4A: Soil Plant Analysis Development (SPAD) reading is not Chlorophyll content. Please revise the Y axis title of Fig. 4A.

Line 164: Fig. 4: “relative conductivity” mentioned in Fig 4 caption is not similar to Fig. 4D. The term should be similar throughout the whole manuscript.

Line 165-169: Please write (A), (B), (C) and (D). after each figure title where applicable.

Line 261: Figure 10. is incomplete. Here the figure did not include antioxidant enzyme, photosynthesis, SPAD/chlorophyll which either decreased oxidative stress or improved growth performance.

Discussion:

Line 290-295: 3.1. Single or joint role of MT and Si in regulating maize growth. The cellular, biochemical and physiological mechanisms of growth promotion and yield increase were not explained. Please discuss with references.

Line 272-287: This part should be at the end of discussion or at the conclusion.

Line 332-334: Single or joint role of MT and Si in alleviating oxidative damage. The process of Oxidative stress generation by Cd was not explained.

Line 362-367: SPAD value and Pn of maize leaves, SOD, POD, and CAT activities. These is no discussion/explanation about these parameters. Please include how MT and Si affect the above mentioned parameters. Include related references/citation too.

Conclusion:

Please mention the results clearly. Mention the limitation of the study and future aspects.

References: Include more related references to support discussion.

Title, Introduction, Discussion should be revised thoroughly and language editing should be done by expert. 

Reviewer 4 Report

Comments to the authors:

After carefully reviewing the article “Mitigation of Cadmium Accumulation and Toxicity in Maize (Zea mays L.) by Joint Application of Melatonin and Silicon”, I have following concerns regarding current manuscript that authors should consider while revising their manuscript:

-Title should be catchy enough, so please revise the title.

-Revise the study objective in the abstract section. In addition, describe brief methodology separately.

-Avoid starting any sentence with the abbreviation, check and correct in the whole manuscript.

-Please add % increase/ decrease of few key parameters in abstract section.

-Please revise the keywords and avoid using the keywords from the title. Moreover, add minimum five keywords.

-Authors should clearly mention that what is the significance and the novelty of current study? Moreover, add research hypothesis at the end of introduction section.

-Add research hypothesis and clear study objective at the end of introduction section.

-Authors should give the details of used protocols for biochemical and physiological parameters.  

-Please provide the names, Manufacturer Company and its location of all used instruments. Moreover, provide the used chemical details as well as quality control and assurance information including the analytical quality control and accuracy.

-Kindly carefully check and correct statistics, especially the lettering on graphs in figure 2. 

-Results are well written and well explained but I recommend adding more detailed mechanistic approach for better understandings.

-I recommend starting the discussion with your own results.

-Please re-write the conclusion in a more comprehensive way.

-References should be according to the journal given format.

Can be improved

Round 2

Reviewer 1 Report

Accept in present form

Accept in present form

Reviewer 2 Report

The manuscript was improved and fit for publicaion